# The Role of Vitamin D and Oxidative Stress in Chronic Kidney Disease

**DOI:** 10.3390/ijerph15122701

**Published:** 2018-11-30

**Authors:** Keith C. Norris, Opeyemi Olabisi, M. Edwina Barnett, Yuan-Xiang Meng, David Martins, Chamberlain Obialo, Jae Eun Lee, Susanne B. Nicholas

**Affiliations:** 1Department of Medicine, David Geffen School of Medicine, University of California Los Angeles, Los Angeles, CA 90024, USA; SuNicholas@mednet.ucla.edu; 2Department of Medicine, Harvard Medical School, Harvard University, Boston, MA 02138, USA; oolabisi@partners.org; 3RCMI Translational Research Network Data Coordinating Center, College of Science, Engineering and Technology, Jackson State University, Jackson, MS 39217, USA; barnett_me@yahoo.com (M.E.B.); jae.e.lee@jsums.edu (J.E.L.); 4Department of Family Medicine, Morehouse School of Medicine, Atlanta, GA 30310, USA; ymeng@msm.edu (Y.-X.M.); cobialo@msm.edu (C.O.); 5Department of Medicine, College of Medicine, Charles R. Drew University of Medicine and Science, Los Angeles, CA 90059, USA; davidmartins@cdrewu.edu

**Keywords:** vitamin D, oxidative stress, kidney disease, disparities

## Abstract

Chronic kidney disease (CKD) is a major non-communicable disease associated with high rates of premature morbidity and mortality. The prevalence of hypovitaminosis D (deficiency of 25(OH)D or 25D) is greater in racial/ethnic minorities and in patients with CKD than the general population. Low 25D is associated with bone and mineral disorders as well as immune, cardiometabolic and cardiovascular (CV) diseases. Thus, it has been suggested that low 25D contributes to the poor outcomes in patients with CKD. The prevalence of hypovitaminosis D rises progressively with advancing severity of kidney disease with over 30% of patients with CKD stage 3 and 70% patients with CKD stage 5 estimated to have low levels of 25D. This report describes several of the abnormal physiologic and counter-regulatory actions related to low 25D in CKD such as those in oxidative stress and inflammatory systems, and some of the preclinical and clinical evidence, or lack thereof, of normalizing serum 25D levels to improve outcomes in patients with CKD, and especially for the high risk subset of racial/ethnic minorities who suffer from higher rates of advanced CKD and hypovitaminosis D.

## 1. Introduction

Chronic kidney disease (CKD) is a major non-communicable disease and is emerging as an important public health problem. Patients with CKD suffer from higher rates of premature morbidity and mortality due to a myriad of metabolic perturbations that arise as renal function declines. The prevalence of vitamin D deficiency (25(OH)D or 25D) is greater in patients with CKD than the general population [1,2,3]. Over 30% of patients with CKD stage 3 (estimated glomerular filtration rate (eGFR) between 30 and 59 mL/min/1.73 m^2^) have low levels of 25D, and the prevalence of 25D deficiency is as high as 60–70% in the later stages of CKD (stage 4 eGFR between 15 and 29 mL/min/1.73 m^2^ and stage 5 eGFR <15 mL/min/1.73 m^2^) [4]. Deficiency of 25D is associated with adverse clinical outcomes such a bone and mineral disorders (BMD) as well as immune, cardiometabolic and cardiovascular (CV) diseases [5,6,7,8]. Thus, improving outcomes in patients with CKD requires normalization of many of the dysregulated physiologic and hormonal systems in CKD. While circulating levels of 25D need to be improved, optimizing the improvement of immune and cardiometabolic health needs to take advantage of the unique interplay between 25D, oxidative stress and inflammation.

## 2. Vitamin D Deficiency/Insufficiency in Chronic Kidney Disease

25D is a pre-hormone that acts in concert with a variety of signaling pathways to influence a variety of cell actions throughout the body [9]. Recommendations for adequate 25D levels vary [10,11], as the optimal concentrations for a given cell function and optimal clinical outcome varies [12]. In addition, the level of 25D at which there is a clear adverse physiological manifestation due to the low value may vary by degree of impairment or activation of related counter-regulatory systems such as oxidative stress and inflammatory pathways [12]. Reasons for 25D deficiency in the CKD population include, but are not limited to, phosphorus-restricted diets as well as reduced dietary intake in general, reduced endogenous synthesis of 25D due to limited outdoor activity and reduced sunlight exposure, and comorbid illnesses [2,13,14,15,16]. In the presence of CKD and other co-morbid conditions, there appears to be an increase in the CYP24A1 enzyme activity which catabolizes both 25D and 1,25D thereby influencing their levels, but more so the levels of their metabolites (Figure 1) [17]. Along those lines, elevated Fibroblast growth factor 23 levels in CKD may suppress 1-alpha hydroxylase gene expression and 1,25 D levels but likely plays no role in modulating 25D levels [17].

Hypovitaminosis D is commonly classified in terms of deficient and insufficient levels but these terms represent a spectrum of disease risk and not an explicit state of disease [9], leading to varying treatment recommendations. The Institute of Medicine (IOM) recommends serum 25D levels be maintained at 20–50 ng/mL [10]. While the optimal serum 25D level for patients with CKD is not well defined, serum 25D concentrations below 12 ng/mL are associated with marked increased risk for BMD, cardiometabolic and cardiovascular diseases [10,11,18]. Different organizations vary in their recommendations of the levels at which hypovitaminosis D occur with 25D levels below 12–20 ng/mL considered deficient and levels below 20–30 ng/mL considered insufficient [10,11].

## 3. Vitamin D, Oxidative Stress and Inflammation

Several signaling pathways are charged with maintaining a healthy balance in the ongoing struggle between injurious oxidant and protective antioxidant events. These pathways include modification of proteins and DNA, and alteration in gene expression that may promote apoptosis, endothelial dysfunction and impairment of cellular immunity [19]. A common metabolic pathway of stress-related cellular activation is an increase in reactive oxygen species (ROS) causing adverse cellular events termed oxidative stress, which is found in many chronic medical conditions such as atherosclerosis, diabetes, immune-related disorders and CKD [19]. Emerging evidence supports the role of 25D administration in attenuating oxidative stress via increased nuclear factor-erythroid-2-related factor 2 (Nrf2) and up-regulation of the expression of genes encoding antioxidant enzymes, as well as modulating levels of ROS through control of cellular antioxidants [20,21,22]. In addition to oxidative stress, inflammation is a second major system implicated in the pathogenesis of the premature CV diseases in patients with CKD [22]. Nrf2 activates the antioxidant response element (ARE) and activation of the ARE downregulates redox-sensitive and inflammatory genes, including nuclear factor-kB (NF-kB) [22]. In patients with CKD, increasing oxidative stress leading to increased inflammation, and vice versa, are part of a deleterious cycle leading to over-production of each and adverse clinical sequelae [22]. The ability of 25D repletion to attenuate this viscous cycle and reduce oxidative stress and inflammation through increasing Nrf2 and activating ARE represents a non-traditional regulatory role of the vitamin D pathway and a potential mechanism through which it may improve CKD-related CV disease, anemia, inflammation, and other clinical disorders [22].

## 4. Select Pre-Clinical and Clinical Studies

While the effects of vitamin D on attenuating oxidative stress in cell cultures and animal models have been robust, clinical results have been mixed, possibly due to factors such as differences in dosing, duration of treatment, and clinical setting (e.g., baseline 25D level, oxidative stress and inflammatory marker levels). Treatment with an analog of 1,25(OH)_2_D, the active vitamin D hormone, reduced markers of systemic and intrarenal oxidative stress in mice with diabetic nephropathy [23]. In addition, pretreatment with 1,25(OH)_2_D administration reduced antioxidant activity and inflammation in a human cell culture model [24] and animal models [25,26] of oxidative stress. Despite promise in preclinical trials, its clinical effect in humans has been more variable. A Cochrane meta-analysis of seven randomized controlled trials of 25D administration to patients with polycystic ovary syndrome and normal kidney function found significant decreases in serum high-sensitivity (hs) C-reactive protein (CRP) and malondialdehyde (a marker of oxidative stress) with increased total antioxidant capacity, but no effect on nitric oxide or total glutathione levels [27]. However, the administration of 1 mcg/day of paricalcitol [19-nor-1,25-(OH)_2_ D_2,_ an analog of the active form of vitamin D_2_] or placebo for three months to 60 patients with diabetes and stage 3 or stage 4 CKD did not affect biomarkers of either oxidative stress or inflammation [28]. By contrast, one month of both 1 and 2 mcg/day of paricalcitol administration to 24 patients with CKD and a mean eGFR of 45 mL/min/1.73 m^2^ significantly reduced hs-CRP and albuminuria compared to placebo [29]. Coyne DW, et al. reported that paricalcitol (1 mcg/day) and 1,25(OH)_2_D (calcitriol; 0.25 mcg/day) were similarly effective in lowering intact parathyroid hormone (iPTH) and alkaline phosphatase in patients with secondary hyperparathyroidism in stages 3–4 CKD, with minimal elevations in serum calcium and phosphorus [30]. However, in regards to clinical outcomes two more recent outcome trials found no benefit from paricalcitol on clinical CV outcomes in patients with CKD. The PRIMO study (Paricalcitol Capsules Benefits in Renal Failure-Induced Cardiac Morbidity) found that forty-eight weeks of therapy with 2 mcg/day paricalcitol vs. placebo did not alter the left ventricular mass index or improve certain measures of diastolic dysfunction in 227 patients with stage 3 and 4 CKD [31]. Similarly, the OPERA trial (Oral Paricalcitol in Retarding Cardiac Hypertrophy, Reducing Inflammation and Atherosclerosis in Stage 3–5 Chronic Kidney Disease) found that a 52-week intervention with 1 mcg/day paricalcitol vs. placebo significantly improved secondary hyperparathyroidism but did not alter measures of left ventricular structure and function in 60 patients with stage 3–5 CKD [32]. Thus, the promise of vitamin D to have a clinically significant impact on CV diseases in patients with CKD has not been shown in recent clinical trials. Whether or not there may be greater efficacy in select patients with higher baseline levels of oxidative stress or inflammation, and if these potential effects differ by dose and type of vitamin D used remains to be determined.

## 5. Possible Implications for Disparities

African Americans/Blacks suffer from overall higher rates of hypovitaminosis D than other racial ethnic groups [33,34,35]. Many cardiometabolic disorders that are also disproportionately high in African Americans are also associated with low levels of vitamin D [33]. In addition, African Americans suffer from higher rates of the advanced stages of CKD and are three times more likely to develop end-stage kidney disease (ESKD) [36]. For patients with CKD, African Americans typically have higher levels of intact parathyroid hormone [32]. Interestingly, compared to White patients, African Americans on dialysis appear to have greater survival which has been linked, in part, to treatment with active vitamin D [1,25(OH)_2_D or analogs] [37,38]. Whether this is an independent effect, related to a vitamin D—oxidative stress/inflammation interaction, or due to other causes is not known.

The role of hypovitaminosis D in African Americans is complex as emerging evidence has suggested that certain vitamin D binding protein polymorphisms are associated with low measured levels of serum 25D but normal bioactive 25D, and these polymorphisms are more prevalent in African Americans [39]. Clinically, this is highlighted by the findings of Gutierrez et al. who reported a strong correlation between serum 25D levels and bone mineral density among White and Hispanic patients, but no correlation for African Americans who had similar bone mineral density measures regardless of serum 25D level [40]. Also, Robinson-Cohen reported increased CV events associated with low serum 25D levels in Whites and Asians from over 6400 participants in the Multi-Ethnic Study of Atherosclerosis, but no relationship for Hispanics and African Americans [41], possibly related to ethnic differences in vitamin D binding protein polymorphisms or the vitamin D receptor [42]. Berg and colleagues found that both 24,25(OH)_2_D levels and 25D levels were higher in White Americans compared to Black Americans, but the ratio of 24,25(OH)_2_D to 25D was the same in both groups [43]. Thus, the 24,25(OH)_2_D to 25D ratio [vitamin D metabolite ratio (VMR)] may reflect free circulating 25D and could represent a more physiologically precise measure of bioactive 25D that should be independent of racial/ethnic differences in vitamin D binding protein levels and/or polymorphisms. Thus, the VMR may represent a new candidate biomarker for vitamin D status. This could mean the lower levels of serum 25D in African Americans may not require treatment and differing rates of these vitamin D binding protein polymorphisms may contribute to the conflicting outcomes in different clinical studies related to vitamin D levels.

However there is still evidence of other racial/ethnic differences in the vitamin D pathways, as Blacks with ESKD have higher levels of iPTH suggesting an increased risk of hyperparathyroid bone disease. Both Wolf et al. and Kalantar and colleagues demonstrated greater rates of survival among Black dialysis patients treated with higher doses of vitamin D analogs compared to either lower doses or no active vitamin D, in contrast to White dialysis patients whose survival rates did not significantly vary with use of active vitamin D analogs [37,38]. FGF23 levels may play a role in the differential vitamin D survival by race/ethnicity as Black patients on dialysis had a 60% lower risk of death vs. White patients among the subset below the population median of FGF-23 levels, while the Black–White mortality rate did not differ in the subset above the population median [44]. Examining the impact of vitamin D on FGF23 levels as a potential mediator or predictor of death is an area of future investigation.

## 6. Conclusions

Observational studies have linked low 25D levels in patients with CKD to progression to end-stage kidney disease, infections, fracture rates, hospitalizations, and all-cause mortality [45]. Unfortunately, while prospective studies of vitamin D analogs in patients with CKD have demonstrated reduced iPTH levels, they have not found improvement in CV outcomes [30,31]. Similarly, a new prospective study of 2000 IU/day of 25D to a general population (no CKD) over 5.3 years showed no difference in CV or cancer outcomes [46]. A recent panel convened by the National Kidney Foundation recommended that patients with CKD and 25D levels less than 15 ng/mL be treated, while those with serum 25D levels between 15 and 20 ng/mL may not require treatment unless there is evidence of counter-regulatory hormone activity [45]. Whether evidence of increased oxidative stress and inflammation are other indicators of increased risk requiring treatment at modestly low serum levels is still to be determined. Additional future studies are warranted to further assess the value of the 24,25(OH)_2_D to 25D ratio or VMR and its correlation with clinical outcomes across racial/ethnic groups overall and in patients with CKD, as well as the effect of 1,25D analogs on FGF23 levels as potential mediators or predictors of CV events and/or death in patients on dialysis as noted earlier.

## Figures and Tables

**Figure 1 ijerph-15-02701-f001:**
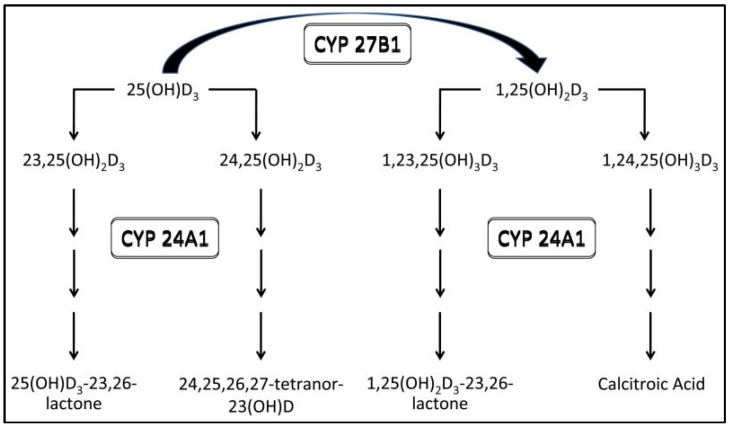
Catabolism of 1,25(OH)_2_D and 25(OH)D. 25(OH)D_3_, 25-hydroxyvitamin D_3_; 1,25(OH)_2_D_3_, 1,25-dihydroxyvitamin D_3_; 23,25(OH)_2_D_3_, 23,25-dihydroxyvitamin D_3_; 24,25(OH)_2_D_3_, 24,25-dihydroxyvitamin D_3_; 1,23,25(OH)_3_D_3_, 1,23,25-trihydroxyvitamin D_3_; 1,24,25(OH)_3_D_3_, 1,24,25-trihydroxyvitamin D_3_; 24,25,26,27-tetranor-23(OH)D, 24,25,26,27-tetranor-23-hydroxyvitamin D [17].

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
