# Peer review of "The Role of Vitamin D and Oxidative Stress in Chronic Kidney Disease"

_ijerph, 2018, doi:10.3390/ijerph15122701_

Round 1

Reviewer 1 Report

The authors of the reviewed Communication have described the role of vitamin D, oxidative stress and inflammation in patients with chronic kidney disease (CKD) as well as in African Americans. They also discussed some pre-clinical and clinical studies when patients were treated with paricalcitol. There are still many concerns whether the treatment with vitamin D analogs can improve some disorders or not, including some CKD-related cardiovascular diseases. The presented topic is interesting and falls into the profile of the International Journal of Environmental Research and Public Health. However, some items listed below should be considered and revised.

1.     What is the advantage of this approach, especially for hypovitaminosis D in racial/ethnic minorities and in patients with CKD, when several reports have been recently published?

2.     What is the novelty factor of this paper?

Author Response

Reviewer 1

The authors of the reviewed Communication have described the role of vitamin D, oxidative stress and inflammation in patients with chronic kidney disease (CKD) as well as in African Americans. They also discussed some pre-clinical and clinical studies when patients were treated with paricalcitol. There are still many concerns whether

the treatment with vitamin D analogs can improve some disorders or not, including some CKD-related cardiovascular diseases. The presented topic is interesting and falls into the profile of the International Journal of Environmental Research and Public Health.

Response

We thank the reviewer for the kind remarks and the constructive queries below.

Question 1

What is the advantage of this approach, especially for hypovitaminosis D in racial/ethnic minorities and in patients with CKD, when several reports have been recently published?

Response

Embedded in response to #2

Question 2. What is the novelty factor of this paper?

Response

To date limited attention has been paid to the relationship of vitamin D and oxidative stress and inflammatory status in patients with CKD (or the general population). The use of 25D and to a greater extent 1,25D and analogs in CKD have shown limited impact on CV and mortality outcomes in prospective trials. Given our emerging understanding of the interaction of Vitamin with oxidative stress and inflammatory pathways and lack of CV outcomes in more general populations, it may be that repletion of vitamin D is more impactful in the setting of adverse oxidative stress and/or inflammatory states. There may also be an interaction with FGF 23 (which differs by race/ethnicity), which is now added. Raising awareness of this issue can lead to more judicious use of vitamin D (beyond PTH reduction but targeting potential patients who may benefit) until additional studies are done.

In addition to mentioning future studies to understand Whether increased oxidative stress and inflammation are other indicators of increased risk would benefit from treatment at modestly low serum levels is still to be determined and to assess the value of the 24,25(OH)2D to 25D ratio and its correlation with clinical outcomes across racial/ethnic groups overall and in patients with CKD…..we have now added: as well as the role 1,25D analogs on FGF23 levels as a potential mediator or predictor of CV events and/or death in patients on dialysis.

Reviewer 2 Report

This conference paper is adressed at an important issue. 

My main concern is that  even if it is well written and synthetic, it does not bring novel views in the overall knowledge on this complex issue nor does it underline the most important novelties.

I have added some suggestions to make the paper more appealing, at least for an average clinical nephrologist like myself.

the link between vit D and kidney disease should be discussed in a more  critical way: the cause effect relationship is far from being clarified; a discussion on what is not known on this issue woud be of interst

the diffferent roles of 25OH vit D and its active an inactive metabolites should be better explaines

the reason why CKD patients display lower 25OH vit D should be discussed

I think that, in the very complex and multifaceted array of elements associated with CKD progression the authors are quite rapid in concluding that the relation with vitamin D is not an epiphenomenon

adding disucssion of what is now new would be intersting

the basis of this review should be highlighted: review in the occasion of a conference, based upon experience... 

suggestions for future research could also be intersting

Author Response

Reveiwer 2

This conference paper is addressed at an important issue.

My main concern is that even if it is well written and synthetic, it does not bring

novel views in the overall knowledge on this complex issue nor does it

underline the most important novelties. I have added some suggestions to make the paper more appealing, at least for an average clinical nephrologist like myself.

Response

We thank the reviewer for the kind remarks and the constructive suggestions below.

Suggestion 1

The link between vit D and kidney disease should be discussed in a more

critical way: the cause effect relationship is far from being clarified; a

discussion on what is not known on this issue would be of interest

the different roles of 25OH vit D and its active and inactive metabolites should

be better explained.

Response

We added comorbid illnesses, and mentioned in the presence of CKD and other co-morbid conditions there appears to be an increase in the CYP24A1 enzyme activity which catabolizes both 25D and 1,25D thereby influencing their levels, but moreso the levels of their metabolites (Bosworth C, de Boer IH. Impaired vitamin D metabolism in CKD. Semin Nephrol. 2013;33(2):158-68). Along those lines, elevated Fibroblast growth factor 23 levels in CKD may suppress 1-alpha hydroxylase gene expression and 1,25 D levels but likely plays no role in modulating 25D levels (Bosworth C, de Boer IH. Impaired vitamin D metabolism in CKD. Semin Nephrol. 2013;33(2):158-68) and a figure from that article

Figure legend: Catabolism of 1,25(OH)2D and 25(OH)D. 25(OH)D3, 25-hydroxyvitamin D3; 1,25(OH)2D3, 1,25-dihydroxyvitamin D3; 23,25(OH)2D3, 23,25-dihydroxyvitamin D3; 24,25(OH)2D3, 24,25-dihydroxyvitamin D3; 1,23,25(OH)3D3, 1,23,25-trihydroxyvitamin D3; 1,24,25(OH)3D3, 1,24,25-trihydroxyvitamin D3; 24,25,26,27-tetranor-23(OH)D, 24,25,26,27-tetranor-23-hydroxyvitamin D.

Suggestion 2

The reason why CKD patients display lower 25OH vit D should be discussed

I think that, in the very complex and multifaceted array of elements associated

with CKD progression the authors are quite rapid in concluding that the relation

with vitamin D is not an epiphenomenon. adding discussion of what is now new would be interesting the basis of this review should be highlighted: review in the occasion of aconference, based upon experience...

Response

in addition to phosphorus-restricted diets as well as decrease in dietary intake in general, reduced endogenous synthesis of 25Ddue to limited outdoor activity and reduced sunlight exposure,we added comorbid illnesses, and mentioned in the presence of CKD and other co-morbid conditions there appears to be an increase in the CYP24A1 enzyme activity which catabolizes both 25D and 1,25D thereby influencing their levels, but moreso the levels of their metabolites (Bosworth C, de Boer IH. Impaired vitamin D metabolism in CKD. Semin Nephrol. 2013;33(2):158-68). Along those lines, elevated Fibroblast growth factor 23 levels in CKD may suppress 1-alpha hydroxylase gene expression and 1,25 D levels but likely plays no role in modulating 25D levels (Bosworth C, de Boer IH. Impaired vitamin D metabolism in CKD. Semin Nephrol. 2013;33(2):158-68).

Suggestion 3

Suggestions for future research could also be interesting

Response

In addition to whether evidence of increased oxidative stress and inflammation are other indicators of increased risk require treatment at modestly low serum levels and future studies to further assess the value of the 24,25(OH)2D to 25D ratio and its correlation with clinical outcomes across racial/ethnic groups overall and in patients with CKD…..

we added: as well as the role 1,25D analogs on FGF23 levels as a potential mediator or predictor of CV events and/or death in patients on dialysis.

Round 2

Reviewer 2 Report

the paper is more critcal in this present form, and I think may be of interest for the readers, in particular since it is adressed not only to a "conventional" nephrology public, but to a wider public health one. 

(just a reminder: the references have to be integrated in the list)